# Renal Biomarkers in Cardiovascular Patients with Acute Kidney Injury: A Case Report and Literature Review

**DOI:** 10.3390/diagnostics13111922

**Published:** 2023-05-31

**Authors:** Rolando Claure-Del Granado, Jonathan S. Chávez-Íñiguez

**Affiliations:** 1Division of Nephrology, Hospital Obrero No 2—CNS, Cochabamba, Bolivia; 2Instituto de Investigaciones Biomédicas e Investigación Social de la Facultad de Medicina (IIBISMED), Facultad de Medicina, Universidad Mayor de San Simon, Cochabamba C.P. 3119, Bolivia; 3Division of Nephrology, Hospital Civil de Guadalajara Fray Antonio Alcalde, Guadalajara C.P. 44280, Mexico; jonarchi_10@hotmail.com; 4University of Guadalajara Health Sciences Center, Guadalajara C.P. 44340, Mexico

**Keywords:** cardiorenal syndromes, acute kidney injury, biomarkers, chronic kidney disease, heart failure

## Abstract

Biomarkers have become important tools in the diagnosis and management of cardiorenal syndrome (CRS), a complex condition characterized by dysfunction in both the cardiovascular and renal systems. Biomarkers can help identify the presence and severity of CRS, predict its progression and outcomes, and facilitate personalized treatment options. Several biomarkers, including natriuretic peptides, troponins, and inflammatory markers, have been extensively studied in CRS, and have shown promising results in improving diagnosis and prognosis. In addition, emerging biomarkers, such as kidney injury molecule-1 and neutrophil gelatinase-associated lipocalin, offer potential for early detection and intervention of CRS. However, the use of biomarkers in CRS is still in its infancy, and further research is needed to establish their utility in routine clinical practice. This review highlights the role of biomarkers in the diagnosis, prognosis, and management of CRS, and discusses their potential as valuable clinical tools for personalized medicine in the future.

## 1. Case Presentation

A 72-year-old man presents to an emergency department with dyspnea and edema that has worsened in the last 4 days; he stopped taking his usual dose of furosemide 2 weeks ago. He has chronic heart failure with reduced ejection fraction and chronic kidney disease (CKD) stage G3aA3 (eGFR of 49 mL/min/1.73 m^2^ by CKD-EPI equation and albuminuria of 346 mg/g). On admission, he presents a SpO_2_ of 89% on room air, a heart rate of 93 beats per minute, tachypnea (26 breaths per minute), afebrile with blood pressure of 146/92 mmHg, plethoric jugular bilateral veins, and pitting edema of the abdominal wall and legs. Lung ultrasound shows 14 B-lines, right pleural effusion of 200 mL, and a Venous Excess Ultrasound Score (VExUS) is grade 3. He presents with worsening renal function with a serum creatinine level of 1.9 mg/dL on admission (baseline serum creatinine level of 1.5 mg/dL), brain natriuretic peptide (BNP) is 32,000 ng/dL, cancer antigen 125 (CA-125) is 134 mg/dL, and the spot albumin/creatinine ratio is 523 mg/g; urinary sodium, 2 h after intravenous administration of 80 mg of furosemide, is 49 mmol/L. The patient is hospitalized in order to promote rapid and effective decongestion.

## 2. Introduction

The relationship between the heart and the kidney is a well-studied interaction in medicine, as any pathology affecting one can have significant consequences for the other. This interaction, known as cardiorenal syndrome (CRS), is prevalent and associated with high morbidity and mortality [1]. There are five types of CRSs, classified based on the initial organ injury causing the syndrome (Table 1) [2]. For instance, Type 1 refers to a patient who experiences an acute myocardial infarction, which further complicates with acute kidney injury (AKI). On the other hand, Type 2 pertains to a patient with left ventricular hypertrophy who develops chronic kidney disease (CKD). Type 3 refers to a patient who initially presents with AKI and subsequently experiences hyperkalemia and cardiac arrhythmias. Type 4 is applicable to a patient undergoing hemodialysis who also manifests congestive heart failure. Lastly, Type 5 encompasses patients who encounter infectious sepsis resulting in myocardial dysfunctions and AKI secondary to the infection [2].

Type 1 cardiorenal syndrome (CRS1) is the most fatal of all, as it involves acute cardiovascular and kidney disorders, leading to an increased risk of mortality up to five times within 28 days of onset. Furthermore, the negative impact of CRS1 continues for up to five years [3,4,5]. The physiopathogenic mechanism of CRS1 is multifactorial, but neurohormonal activation, subclinical inflammation, and vascular congestion prevail [1]; these factors typically contribute to the development of nephrosarca, causing sodium and water retention [6]. The diagnosis of CRS1 is made through various clinical and laboratory tests, including ultrasound and natriuretic peptide and CA-125 measurements. Due to the physiopathogenic mechanism of CRS1, the trajectory and trends of serum creatinine levels and estimated glomerular filtration rate (eGFR) may not always be accurate and lack clinical precision. This is primarily due to the limitations of using creatinine as a biomarker in critically ill patients, who often experience low rates of creatinine generation in inflammatory states, and due to malnutrition, muscle catabolism, and drugs that compete against tubular creatinine secretion [7]. Therefore, researchers have worked to identify new kidney and cardiac biomarkers that may predict adverse outcomes, such as hospitalization or death, and improve kidney replacement therapy (KRT) timing and efficacy [8] (Figure 1). Overall, using biomarkers other than creatinine and urinary output as part of an AKI classification would help identify high-risk patients early and improve outcomes.

The main goal is to implement a new AKI classification which includes biomarkers other than serum creatinine and urinary output. This would allow timely identification of susceptible patients and ultimately enhance patient outcomes [9].

## 3. Biomarkers for Diagnosing AKI during Cardiac Failure

Although serum creatinine has its shortcomings and constraints, it remains a critical biomarker for patients with cardiovascular ailments. A prime illustration of its significance is the AKINESIS multinational cohort study, which examined 787 patients with acute heart failure (AHF) and measured five biomarkers, including brain natriuretic peptide, high sensitivity cardiac troponin I, galectin 3, serum neutrophil gelatinase-associated lipocalin, and urine neutrophil gelatinase-associated lipocalin, upon hospital admission The findings revealed that serum creatinine is the most reliable predictor of declining renal function [10].

These findings stand in contrast to other studies where a new biomarker (Cystatin C) was found to be a more effective predictor than serum creatinine. Among 262 patients with AHF, stratifying the risk of death and re-hospitalization, based on either serum Cystatin C or serum creatinine values, revealed that Cystatin C was superior at predicting which patients would develop these complications, with an area under the curve (AUC) of 0.66 compared to 0.60 for serum creatinine (*p* ≤ 0.05) [11].

Albuminuria reflects damage to the kidney filtration barrier, vascular endothelial dysfunction, systemic inflammation, tubular damage, and neurohormonal activation. It has been associated with worsening fluid overload through increases in sodium and water retention and cardiac filling pressures, ultimately exacerbating heart failure [12]. Furthermore, albuminuria serves as an excellent risk marker for the development of AKI, with higher values indicating greater risk and renal susceptibility [13].

The natriuretic peptides have utility in AKI patients. Salah et al. analyzed dynamic changes in kidney function and N-terminal pro-B-type natriuretic peptide (NT-proBNP) in 1232 patients hospitalized for AHF. They observed that a reduction in NT-proBNP of more than 30% during hospitalization was associated with worsening kidney function, but lower rates of mortality, independent of the presence of AKI [14]. This highlights the importance of promoting effective decongestion in patients with CRS1 and vascular congestion, regardless of AKI status. Proenkephalin A (PENK), an endogenous opioid with a crucial role in cardiovascular regulation, has been linked to worsening kidney function in AHF patients. Patients with PENK values >147 pmol/L had a 58% increased probability of worsening kidney function, and PENK levels also predicted death at one year [15].

Neutrophil gelatinase-associated lipocalin (NGAL) is a well-researched biomarker, but has limitations in predicting acute kidney injury during hospitalization. A study of 207 AHF patients found that plasma NGAL values >114 ng/mL were associated with a 340% increased risk of developing AKI but had an AUC of 0.67, performing poorly [16]. Similarly, a prospective cohort of 60 AHF patients found that baseline NGAL values had an AUC of 0.67, significantly worse than serum creatinine with an AUC of 0.69 [16].

High levels of a renin–angiotensin–aldosterone system in urine and systemically, due to the intense neurohormonal activation of CRS, may have implications in predicting AKI in AHF patients. Urinary angiotensinogen (AgtU) values were studied in a multicentric cohort of 732 AHF patients, where it was observed that elevated AgtU values were associated with a 10-fold increase in the probability of complications with AKI grades 1–2 [17]. Furthermore, AgtU performed much better (AUC 0.78) than urinary NGAL, interleukin-18 (IL-18), and clinical models, with AUCs of 0.74, 0.73, and 0.77, respectively [17]. Therefore, AgtU may be a more reliable predictor of AKI in AHF patients than NGAL.

Serum sodium levels are closely linked to both neurohormonal activation and water balance. Thus, it is plausible that they play a crucial role in predicting declining kidney function among those with AHF. This theory is supported by the analysis of two cohorts of over 3000 patients with AHF and acute myocardial infarction (AMI), which found that hyponatremia (serum sodium < 136 mmol/L) was associated with a 90% and 50% increase in AKI risk among AHF and AMI patients, respectively [18].

Hospitalized patients with AHF undergoing decongestion frequently develop AKI, with nearly half affected [19]. Although minor increases in serum creatinine of <0.5 mg/dL are permissible during treatment, it is possible that more significant increases could be beneficial in preventing medium and long-term cardiorenal events. Ultimately, the priority should be on achieving effective decongestion, even if this results in modest deterioration of kidney function [20].

## 4. Predicting AKI in Cardiac Surgery

In cardiac surgery patients, AKI occurs in 25–50% of cases [21]. Predicting who will develop this syndrome before surgery can improve clinical management and prognosis. Many kidney and cardiac biomarkers have been explored in this setting because the moment of renal insult is known during cardiac surgery. Certain biomarkers, even when serum creatinine remains normal, have prognostic value and predict increased mortality risk [22]. In the following section, we discuss the different biomarkers used to predict AKI in cardiac surgery, ranging from the most affordable and available to the most novel and experimental.

Urinary output is an important metric, and a study of 6637 patients undergoing cardiac surgery showed that those with oliguria (<400 mL/day) had a 7.6% major adverse kidney event (MAKE) within 90–180 days, compared to 4.5% of those who did not have oliguria [23].

Rapid increases in serum creatinine in the first few hours of surgery can help predict the development of AKI in the following days. In a prospective study of 350 patients, a percentage change of >10% from the basal value was associated with up to a six-fold increase in the probability of developing AKI [21].

Proteinuria and albuminuria, detected by urine test strips, are associated with an increased risk of acute kidney injury (AKI) in patients prior to cardiac surgery. Among 1198 evaluated patients, those with the highest proteinuria values demonstrated a three-fold increase in risk, while those with high albuminuria values had a 66% higher risk [24]. Uric acid has historically been linked to kidney function, and a prospective cohort of 247 patients undergoing elective cardiac surgery revealed that uric acid values greater than 6.3 mg/dL were associated with a fivefold increase in AKI risk [25].

Functional renal reserve (FRR) is another potential biomarker that measures the kidney’s ability to increase its glomerular filtration rate in the face of physiological stress. A normal FRR increase is considered to be over 30 mL/min before the stimulus. In a cohort of 110 patients undergoing elective cardiac surgery, poor FRR (<10 mL/min) was present in all patients who developed AKI, compared to none of those with an FRR greater than 40 mL/min, with an AUC of 0.83 [26].

Cystatin C is a highly available biomarker and one of the most commonly used [27]. A prospective cohort of 412 patients in a coronary care unit found that Cystatin C values > 2.86 mg/L on admission were associated with a nine-fold increase in AKI risk, as well as lower survival rates and higher readmission rates [28]. A meta-analysis of 28 studies confirmed these findings, showing that measuring plasma Cystatin C values within the first 24 h of cardiac surgery demonstrated predictive value, with an AUC of 0.69 [29]. When added to existing clinical scores such as EuroSCORE or congestive cardiac failure (CCF) score, it improved significantly, with AUC values of 0.80 and 0.74, respectively [30].

In a cohort of 26 patients undergoing cardiac surgery, renal hypoxia was measured by urinary oxygen tension (PO_2_). The study observed that patients with PO_2_ > 15 mmHg for more than 4.8 min during the surgery had a five-fold increase in the risk of developing AKI [31].

The Nephrocheck biomarker has been extensively studied, and its usefulness demonstrated even in patients with multiple comorbidities, such as chronic heart failure, with an AUC close to 0.88 [32,33]. In a study conducted in a single center, 57 patients who underwent cardiac surgery were observed. Out of these, 20 (35%) developed AKI. The receiver operating characteristic curve (ROC) of [Tissue inhibitor of metalloproteinases-2 (TIMP-2)] × [insulin-like growth factor-binding protein 7 (IGFBP7)] measured at 4 h after admission to the intensive care unit (ICU) was 0.80 (95% CI: 0.68–0.91) for AKI development and 0.83 (95% CI: 0.69–0.96) for stage 2–3 AKI. The urinary [TIMP-2] × [IGFBP7] values at 4 h after ICU admission were significantly higher in patients who developed AKI compared to those who did not (*p* < 0.001) [34].

Growth-differentiation factor-15 (GDF-15) is a promising humoral marker for risk stratification in cardiovascular disease. In 1176 patients undergoing elective cardiac surgery, elevated GDF-15 values were associated with AKI stage 3. The study reported an AUC of 0.84 [35].

Urinary matrix metalloproteinase-7 (uMMP-7) is another novel biomarker that reflects the activity of intrarenal Wnt/b-catenin, activated in AKI models. In a multicenter cohort of >700 adults and children undergoing cardiac surgery, elevated uMMP-7 values in the first hours were associated with AKI. The study found an AUC > 0.76, outperforming other biomarkers and clinical scores [36].

A meta-analysis evaluated the predictive capacity of different biomarkers (Kidney injury molecule 1 (KIM1), NGAL, liver-type fatty acid-binding protein (L-FABP), and Cystatin C) in the development of AKI after cardiac surgery. Biomarkers measured during surgery had an AUC < 0.75, and those measured in the postoperative period performed even worse, with an AUC < 0.72 [29]. However, another study observed that incorporating elevated plasma NGAL and Cystatin C values into clinical scores improved their performance in predicting AKI considerably, with an AUC > 0.80 for both [30].

## 5. Biomarkers for Assessing CRS Pathophysiology

The definition of CRS, known as the “five-subtype” approach, has been described in detail in Table 1. This approach segments CRSs based on the initiator and target organ, as well as the disease onset (acute/chronic). While this approach brought attention to the disease, it hindered the exploration of CRS pathophysiology, leading to minimal impact on clinical management [2,37]. Biomarkers in CRS can be grouped into biomarkers of function, which reflect glomerular filtration and integrity, biomarkers of damage, and biomarkers of cell-cycle arrest (Table 2).

One example of a biomarker of function is albumin, which not only shows glomerular injury but also endothelial dysfunction. Plasma Cystatin-C, which is not affected by muscle mass but by volume status, can differentiate AKI caused by diuretic-induced volume depletion or parenchymal injury. The Renal Optimization Strategies Evaluation–Acute Heart Failure (ROSE-HF) trial demonstrated that protocol-driven aggressive diuresis caused a decrease in eGFR using Cystatin C in 21% of participants, but it was not associated with an increase in kidney tubular injury markers [47]. On the other hand, urinary NGAL showed diagnostic utility in differentiating true AKI from pseudo-worsening kidney function [41].

Biomarkers of function, damage, and cell-cycle arrest, while useful in diagnosing AKI and/or cardiac events, are challenging to implement. To improve diagnostic accuracy and discriminate AKI etiology, the 23rd Acute Disease Quality Initiative suggests combining damage and functional biomarkers to identify high-risk patients [9]. In CRS, biomarkers can aid in understanding heart–kidney injury. Kidney damage biomarkers (e.g., plasma/urinary NGAL, urinary IL-18, urinary KIM-1, urinary L-FABP, urinary NAG, and urinary angiotensinogen) and cell-cycle arrest biomarkers (e.g., urinary [TIMP2] × [IGFBP7]) can inform in CRS Type 1, while cardiac damage biomarkers (e.g., troponin and BNP) can inform in CRS Type 3 and 4. For CRS Type 2 and 5, more study is required, although plasma Cystatin-C, proenkephalin A, and plasma galectin-3 show potential for CRS Type 2 and 4, respectively. CRS Type 5 likely requires a combination of biomarkers since it represents simultaneous heart and kidney injury [46].

## 6. Guiding Treatment with Biomarkers

First-line treatment for fluid removal in HF remains loop diuretics, but there is uncertainty regarding how to administer them [48]. Symptoms and signs used to assess congestion have moderate specificity and low sensitivity, making it important to use biomarkers and imaging techniques to guide treatment. A glycoprotein, synthesized by coelomic epithelial cells in places such as the pleura, pericardium, and peritoneum, called carbohydrate antigen 125 (CA125) [49], traditionally used for ovarian cancer monitoring, has been identified as elevated in hydropic states such as HF [50]. Recent evidence shows the association of CA125 with systemic congestion parameters and surrogate biomarkers of inflammation and congestion [51]. Two clinical trials have evaluated a diuretic strategy guided by plasma concentrations of CA125 versus standard of care, resulting in significant reductions in composite endpoints and improved kidney function. The first trial involved assigning 380 patients, who were discharged with acute HF and high CA125 levels, to either a CA125 strategy or standard care. The goal was to reduce CA125 levels to ≤35 U/mL by modifying diuretic doses, administering statins, and closely monitoring patients. The CA125 strategy resulted in a significant reduction in the primary endpoint of a composite of death or acute HF readmission within one year [52]. In the second trial, 160 patients with acute HF and kidney injury were randomly assigned to receive loop diuretic doses based on CA125 levels or clinical evaluation. The CA125-guided group had higher furosemide equivalent dosages and urinary output over 72 h, and significantly improved eGFR [53].

Natriuretic peptide (NP) levels are of significant prognostic value in various clinical settings. These peptides are released by the heart in response to pressure and volume overload. B-Type natriuretic peptide (BNP) and N-terminal-proBNP have become essential diagnostic tools for evaluating patients who present with acute dyspnea. The NP level indicates the overall function of the heart, including systolic and diastolic function, as well as right ventricular and valvular function [54]. When treating decompensated heart failure patients experiencing volume overload, a decrease in wedge pressure often leads to a rapid decrease in NP levels, which can help with hemodynamic assessment and subsequent treatment adjustment. Monitoring NP levels in the outpatient setting may also improve patient outcomes. In these cases, a reduction in wedge pressure will typically result in an immediate decrease in NP levels within the first 24 h, as long as proper urine output is maintained [55].

To effectively monitor inpatients with decompensated HF, it is reasonable to obtain an NP level upon admission and just prior to discharge, when the patient is believed to be euvolemic. These levels can aid in predicting outcomes and determining the appropriate level of post-discharge care. It may also be helpful in establishing the patient’s “dry weight” NP level, although additional diuresis after discharge may result in a lower NP level [56]. Additional NP level measurements may be needed if there are significant changes in the patient’s condition during hospitalization, although the benefits of this approach have not been studied.

In the diagnosis, therapy, and prognosis of CRS, multi-biomarker strategies may be essential for optimizing clinical outcomes. The combination of biomarkers can increase their accuracy, but it is necessary to determine the optimal combinations to use [57].

## 7. Predicting Outcomes with Biomarkers

Various biomarkers that indicate function, damage, or cell-cycle arrest may predict different outcomes for CRS.

### 7.1. Biomarkers of Function

A study involving three large chronic HF trials showed that microalbuminuria (30–299 mg/g) and macroalbuminuria (≥300 mg/g) are associated with a 1.4–1.8-fold increased risk of all-cause death, cardiovascular death, or HF hospitalization [38,39,40]. Another study revealed that, plasma Cystatin-C can moderately predict AKI (AUC-ROC 0.68) and all-cause death or HF hospitalization (AUC-ROC, 95% confidence interval: 0.73, 0.66–0.80), in acute HF patients [42,43]. While NT-proBNP was better than galectin-3 for diagnosis in acute HF, galectin-3 may be superior to NT-proBNP in predicting 60-day mortality (AUC-ROC 0.74 vs. 0.67, *p* = 0.05), and was associated with a 14-fold increased risk of all-cause death or HF hospitalization in multivariate analysis [58]. Additionally, in acute HF, proenkephalin A can also moderately predict AKI (AUC-ROC 0.69), and was independently linked to a 27% increased risk of 1-year mortality or HF hospitalization [15].

### 7.2. Biomarkers of Damage

Plasma and urinary NGAL were linked with a 1.3–2-fold rise in long-term mortality risk in acute HF [41,44]. KIM-1 is a glycoprotein expressed by renal proximal tubule epithelium in response to injury. In patients with clinical AKI, KIM-1 provided additional prognostic information for 3-year mortality risk, and the highest tertiles showed adjusted hazard ratios ranging from 2.0 to 3.2 [22]. Urinary IL-18, a pro-inflammatory cytokine produced in response to tissue injury, modestly predicted AKI-to-CKD transition at 6 months (AUC-ROC 0.674, 0.543–0.805) [45], and was linked with a modest 1.2-fold increase in long-term mortality risk [22]. While in a prospective multicenter study, uAGT in patients with AKI stage 1 or 2 predicted AKI worsening and AKI progression with subsequent death (AUC-ROC 0.76, 95% CI 0.46 to 1.06; and AUC-ROC 0.93, 95% CI 0.50 to 1.36, respectively) [17].

## 8. Back to the Case

The patient was diagnosed with Type 1 cardiorenal syndrome, with severe vascular congestion promoting AKI. Based on their initial low urinary sodium levels, it was predicted that an increase in the dose of loop diuretics and multisegmental blockade of the nephron with thiazides, acetazolamide, and iSGLT2 would be required. Over the first 24 h, the furosemide dose was increased to 240 mg/day, and 500 mg of acetazolamide was administered orally, resulting in a daily urinary volume of 4200 mL. Over 24, 48, and 72 h, urinary sodium levels improved steadily, measuring at 233, 192, and 160 mmol/L, respectively. BNP levels decreased to 5120 ng/dL, pulmonary B-lines decreased to 2, and the VexUS score was 0. After three months, the patient reported a return to normal life, including physical activity, and held a steady adherence to their treatment routine. His kidney function matched that prior to hospitalization during his programmed clinic visit.

## 9. Conclusions

The early and accurate identification of vulnerable patients who may develop AKI in conjunction with cardiovascular diseases is paramount to effective intervention and mitigating the risk of serious complications. Although serum creatinine has been widely used as a diagnostic method, its limitations have become increasingly evident, prompting the need to expand our diagnostic tools. The use of biomarkers in cardiorenal syndrome and acute kidney injury has revolutionized the diagnosis and timely intervention of these conditions. Biomarkers have allowed for earlier and more accurate diagnosis, improved risk stratification, and targeted therapy. Several biomarkers have been identified, validated, and incorporated into clinical practice, and their usefulness in predicting clinical outcomes is increasingly being realized. However, the interpretation of biomarkers still requires clinical judgment and additional studies are needed to further validate their utility. Nevertheless, the use of biomarkers holds tremendous promise for improving patient outcomes and reducing the burden of cardiorenal syndrome and acute kidney injury.

## Figures and Tables

**Figure 1 diagnostics-13-01922-f001:**
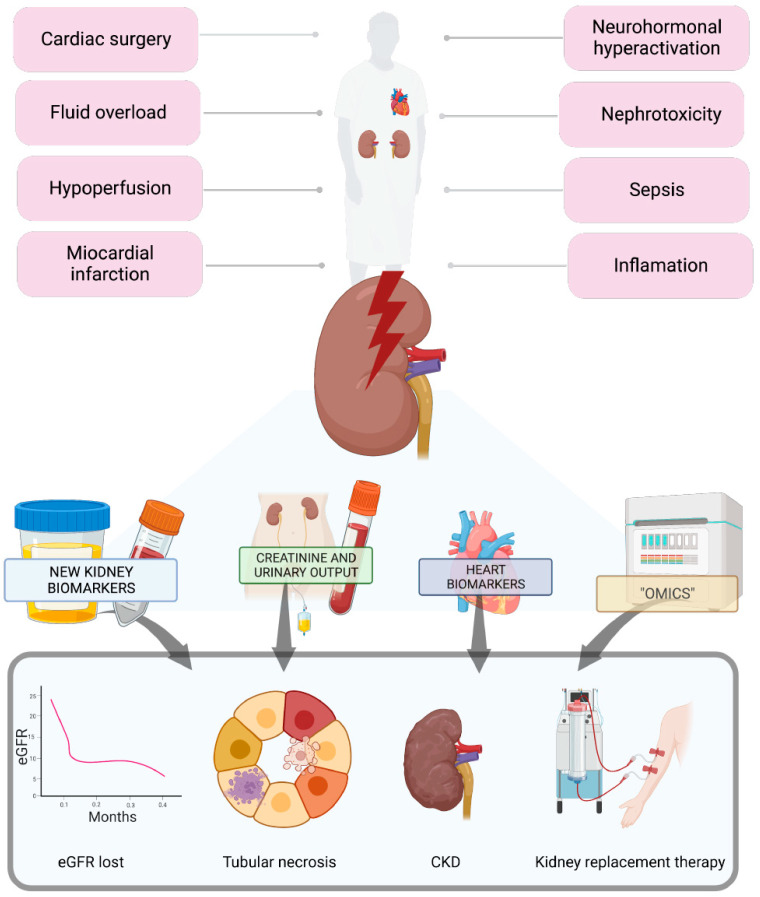
Patients with cardiorenal syndromes are more prone to developing acute kidney injury, and their ability to respond to physiological stressors is limited. To detect acute kidney injury early and predict its progression, various biomarkers have been studied based on the pathophysiological understanding of the syndrome.

**Table 1 diagnostics-13-01922-t001:** Cardiorenal syndrome subtypes.

Cardiorenal Subtype	Description	Examples/Etiology
CRS Type 1 (acute CRS)	Rapid worsening of cardiac function leading to acute kidney injury	Acute MI with cardiogenic shock, ADSHF, acute valvular insufficiency
CRS Type 2 (chronic CRS)	Chronic abnormalities in cardiac function leading to chronic kidney disease	Chronic inflammation, long-term RAAS and SNS activation, chronic hypoperfusion
CRS Type 3 (acute renocardiac syndrome)	Acute worsening of renal function leading to cardiac dysfunction (HF, arrhythmia, and so forth)	Uremia causing impaired contractility, hyperkalemia causing arrhythmias, volume overload causing pulmonary edema
CRS Type 4 (chronic renocardiac syndrome)	Chronic worsening of renal function leading to worsening cardiac function	CKD leading to LVH, coronary disease and calcification, diastolic dysfunction, and so forth
CRS Type 5	Acute or chronic systemic disease leading to both cardiac and renal dysfunction	Diabetes mellitus, amyloidosis, sepsis, vasculitis

CRS, cardiorenal syndrome; MI, myocardial infraction; ADSHF, acute decompensated systolic heart failure; RAAS, renin-angiotensin-aldosterone system; SNS, sympathetic nervous system; HF, heart failure; CKD, chronic kidney disease; LVH, left ventricular hypertrophy.

**Table 2 diagnostics-13-01922-t002:** Biomarkers in cardiorenal syndromes.

	Function of Biomarker	Predictive Value (AUC-ROC)	Prognostic Value (Increase Times Risk of Outcome)	References
**Biomarkers of function**				
Albuminuria	Marker of glomerular injury	Unclear	Type 2 CRS: all-cause/CV death or HF hospitalization (1.4–1.8 times)	[38,39,40]
Plasma Cystatin-C	Produced by all nucleated cells, marker of eGFR	Type 1 CRS: AKI (0.68), all-cause death or hospitalization (0.73)	Type 1 and 2 CRS: all-cause death (2–3 times)	[41,42,43]
Plasma proenkephalin A	Involved in opiod receptor-mediated negative inotropic effects; inversely related to eGFR	Type 1 CRS: AKI (0.69)	Type 1 CRS: all-cause death or HF hospitalization (1.3 times)	[15]
**Biomarkers of kidney damage**				
Plasma and/or urinary NGAL	Secreted by neutrophils and epithelial cells in response to inflammation. Mediates cardiac fibrosis by aldosterone	Type 1 CRS: AKI (0.775–0.996)	Type 1 CRS: all-cause death (1.3–2 times); AKI (5 times)	[17,41,44]
Urinary KIM-1	Facilitates phagocytosis of apoptotic renal tubular cells	Type 1 CRS: AKI (0.83–0.88)	Type 1 CRS: all-cause death (2 times)Type 2 CRS: all-cause death or HF hospitalizations (1.1–1.5 times)	[22,41,45]
Urinary IL-18	Marker of injury from NLRP3-inflammasome on cardiac myocytes and renal tubular cells	Type 1 CRS: AKI (0.61–0.75); AKI to CKD (0.674)	Type 1 CRS: AKI (3.6 times); all-cause death (1.2 times)	[17,22,45]
Urinary L-FABP	Binds fatty acid oxidation products	Type 1 CRS: AKI (0.86) when combined with NAG	Unclear	[29]
Urinary NAG	Renal proximal tubule brush border marker	Unclear	Type 2 CRS: all-cause death (1.3–1.4 times); HF hospitalizations (1.2 times)	[46]
Urinary angiotensinogen	Marker of intrarenal RAAS activation	Type 1 CRS: AKI (0.78); all-cause death (0.86)	Unclear	[17]
**Cell-cycle arrest biomarkers and other biomarkers**				
Urinary (TIMP2) × (IGFBP7)	Involved in G1 cell-cycle arrest during early phase of cell injury	Type 1 CRS: AKI (0.75–0.84)	Unclear	[32,46]
Plasma sydecan-1	Marker of glycocalyx injury	Type 1 CRS: AKI (0.741); severe AKI (0.812); all-cause death (0.788)	Type 1 CRS: all-cause death (1.3 times)	[46]

CRS, cardiorenal syndrome; CV, cardiovascular; AUC-ROC, area under the receiver operating characteristic curve; RAAS, renin–angiotensin–aldosterone system; GFR, glomerular filtration rate; HF, heart failure; CKD, chronic kidney disease; AKI, acute kidney injury; NGAL, neutrophil gelatinase-associated lipocalin; KIM-1, kidney injury molecule-1; L-FABP, liver fatty acid-binding protein; NAG, *N*-acetyl-beta d-glucosaminidase; IL, interleukine; TIMP2, tissue inhibitor of metalloproteinases-2; IGFBP7, insulin-like growth factor-binding protein-7.

## Data Availability

The participant of this case report did not give written consent for her/his data to be shared publicly, so due to the sensitive nature of the data is not available.

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
