# Peer review of "Renal Biomarkers in Cardiovascular Patients with Acute Kidney Injury: A Case Report and Literature Review"

_diagnostics, 2023, doi:10.3390/diagnostics13111922_

Round 1

Reviewer 1 Report

In the present study, the authors have treated a 72-year-old man affected with Type 1 cardiorenal syndrome (CRS), with severe vascular congestion promoting acute kidney injury. In addition, they have reviewed the role of biomarkers in the diagnosis, prognosis, and management of CRS. The manuscript is well written and composed. I have minor comments.

1.       The authors are suggested to add their case report in title.

2.       As the authors have used their patient as a case report, were they taken approval prior to write his case report.

3.       Was the study approved by ethical review board, if yes, please add it in case presentation.

4.       At some points, the authors have used abbreviations before writing full words, e.g. BNP (Case presentation, page 1), CA-125, (Introduction, page 2) etc.

5.        Rearrange Table 2. Table . cont…. ?

Author Response

We thank the reviewer for her/his comments, bellow please find a point by point response.

  1. We have modified the title to: Renal Biomarkers in Cardiovascular Patients with Acute Kidney Injury: a case report review.
  2. Yes, we have a consent form for writing this report, we have attached the signed consent form.
  3. The use of patient information for this case report review was approved by the IRB, we have added a declaration statement at the end of the manuscript.  
  4. Thank you for pointing this. Full words of all abbreviations are now written before them. 
  5. Table 2 is now rearrange 

Reviewer 2 Report

The subject of biomarker use in identification of patients with CRS is extensively evaluated by scientists, but the number of parameters that can be transferred from theory into clinical reality, is still limited. This review fills in this gap and describes in detail the potential of multiple biomarkers versus everyday practice. It is well written and easy to follow, showing step by step various possibilities of diagnosing, differentiating and predicting the patients' risk for AKI development or death.

In cntrats, current conclusions seem very general and non-practical, and as such, should be rephrased into more precise instructions on which parameters are useful in which clinical conditions.

Minor:

The results shown in Table 2 (eg. AUC-ROC values) lack references.

In the description of ref.10, the Authors state that 5 biomarkers, out of which creatinine was the most accurate, were tested. Please name them. 

Author Response

We thank for the reviewer comments/suggestions who have improved our manuscript, bellow please find a point by point response.

  1. The conclusions section was rephrase as suggested.
  2. References were added to table 2 as recommended.
  3. The five biomarkers measured in the study of reference number 10 are now mentioned in the text as requested.